# The structure and robustness of ecological networks with two interaction types

**Virginia Domínguez-García**[1,2]*, **Sonia Kéfi**[1,3]

**1** ISEM, Univ Montpellier, CNRS, IRD, Montpellier, France, **2** Estación Biológica de Doñana (EBD-CSIC), Seville, Spain, **3** Santa Fe Institute, Santa Fe, New Mexico, United States of America

* virginia.dominguez@ebd.csic.es

## Abstract

Until recently, most ecological network analyses investigating the effects of species' declines and extinctions have focused on a single type of interaction (e.g. feeding). In nature, however, diverse interactions co-occur, each of them forming a layer of a 'multilayer' network. Data including information on multiple interaction types has recently started to emerge, giving us the opportunity to have a first glance at possible commonalities in the structure of these networks. We studied the structural features of 44 tripartite ecological networks from the literature, each composed of two layers of interactions (e.g. herbivory and pollination), and investigated their robustness to species losses. Considering two interactions simultaneously, we found that the robustness of the whole community is a combination of the robustness of the two ecological networks composing it. The way in which the layers of interactions are connected to each other affects the interdependence of their robustness. In many networks, this interdependence is low, suggesting that restoration efforts would not automatically propagate through the whole community. Our results highlight the importance of considering multiple interactions simultaneously to better gauge the robustness of ecological communities to species loss and to more reliably identify key species that are important for the persistence of ecological communities.

## Author summary

In the face of the current biodiversity crisis, predicting how species loss will affect ecological communities is becoming increasingly relevant. Previous studies including only one type of ecological interactions (e.g. feeding or pollination) revealed the relevance of the structure of ecological networks for the persistence of ecological communities. However, there is mounting evidence that considering multiple interactions simultaneously can alter the results based on a single interaction. Here, we study the robustness of ecological networks with two interaction types to the loss of plant species, and we show that it is a combination of the robustness of the two bipartite ecological networks composing the ecological community. By analyzing networks from multiple communities, we are able to identify commonalities across interaction types, as well as singularities specific to a given interaction type, caused by underlying biological constraints. Our results highlight that a

**Data Availability Statement:** The data and code supporting the results is available in Zenodo: https://doi.org/10.5281/zenodo.10198613.

**Funding:** Authors acknowledge funding from ANR-18-CE02-0010, EcoNet (Advanced statistical

modelling of ecological networks) of the French National Research Agency (ANR) awarded to SK, that funded VDG. The funders had no role in study design, data collection and analysis, decision to publish, or preparation of the manuscript.

**Competing interests:** The authors have declared that no competing interests exist.

multi-interaction approach is crucial to better gauge the overall robustness of ecological communities, and to correctly determine the relative importance of different plants species at the whole community level, which can be key for biodiversity conservation.

## Introduction

The rate of decline of many species populations is accelerating [1], and species extinctions are seriously threatening the functioning of ecological communities worldwide. Understanding how species interact and how this affects the robustness of ecological communities to species loss is essential to anticipate the consequences of biodiversity losses and extinction cascades as well as to design protection and restoration plans. The study of ecological networks—where species are represented by nodes and the ecological interactions by links between these nodes —have contributed significantly to the understanding of how ecological interactions are structured and have unveiled important relationships between network structure and their robustness to species loss [2–5]. However, while the ecological network literature has long been dominated by studies of networks containing a single interaction type, it has become increasingly clear that species in nature are connected by a myriad of interaction types simultaneously and that considering networks which include this diversity of interaction types could greatly improve our knowledge of the structure and dynamics of ecological communities [6–13].

A number of previous studies have investigated the effect of including multiple interaction types on the functioning of ecological communities, especially on their stability [14–20]. Yet the vast majority of these studies have so far remained theoretical. With the publication of the first multi-interaction empirical networks, we begin to know more about their structure [6, 10, 13, 21–27], and how this structure affects their persistence [6, 10] and robustness [21, 24, 27, 28]. In particular, studies on multi-interaction networks have provided new insights on whether the inclusion of several interactions can significantly alter their robustness to species loss [24] and how extinctions propagate through such networks [21]. However, in spite of these pioneering studies, there is currently no consensus about the structure of multi-interaction networks and its consequences for the robustness of ecological communities, in part due to the lack of data sets, whose amount has only recently started to increase.

A key question, of relevance given the current biodiversity crisis, is how robustness varies across network types, and what we can learn from including multiple interactions simultaneously. With this in mind, we gathered ecological networks with multiple interaction types currently available in the literature. More specifically, we focused on tripartite networks because they were the most abundant in the literature, allowing us to compare a wide variety of ecological systems. Tripartite ecological networks are composed of two interaction layers (e.g. pollination and herbivory), each of the bipartite kind [29]. They therefore contain three different *species sets* (e.g. plant, pollinator and herbivore guilds in a pollination-herbivory network), one of which is shared between the two interaction layers (e.g. plant species can interact with both pollinators and herbivores in a pollination-herbivory network). We call the set of nodes that can have interactions in both interaction layers the *shared set*, and the subset of nodes in the shared set that have interactions in both interaction layers the *connector nodes* (see Fig 1A and 1B).

Our data set consists of 44 tripartite networks from 6 different studies, in which the interaction layers include mutualistic (pollination, seed-dispersal and ant-mutualism) and antagonistic (herbivory and parasitism) interactions (see Methods). To identify possible generalities across interaction types as well as singularities specific to a given interaction type, we divided

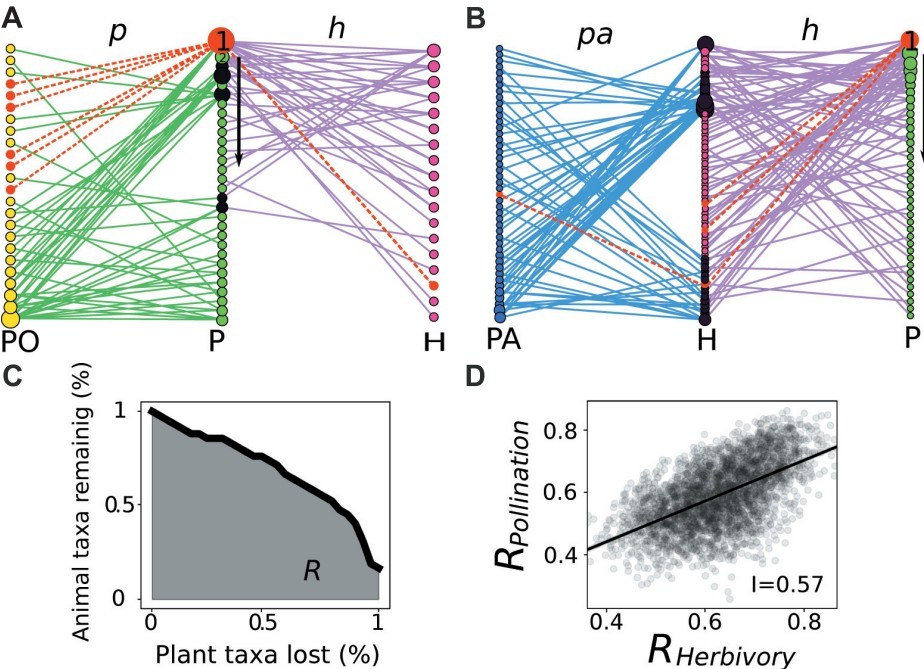

**Fig 1. Tripartite networks, robustness and interdependence.** A) An Herbivory(h)—Pollination(p) tripartite network, where plants (P) are the shared set of species. B) An Herbivory(h)—Parasitism(pa) tripartite network, where herbivores (H) are the shared set of species. Link colours represent the two interaction layers, and node colours the three sets of species. Connector nodes in the shared set of species are highlighted in black. C) Extinction curve showing the fraction of surviving animal species as a function of plant loss for a given plant extinction sequence in network A. The robustness to plant loss, $R$, is the area under the curve. Extinction protocol: plants (green nodes) are progressively removed from the community in the prescribed order, their corresponding links are erased (colored in red) and animal species are declared extinct (colored in red) whenever they lose all their feeding links. D) Pairwise correlation in the robustness of the two animal sets—interdependence, $I$—resulting from 3.000 simulations of random sequential loss of plant taxa in network A.

the networks in three types according to the signs of the interactions involved: mutualism-mutualism (MM) if both interactions were positive, antagonism-antagonism (AA) if both interactions were negative, and mutualism-antagonism (MA) if one interaction was positive and the other negative, given that interaction type can determine network architecture through the underlying biological constraints [30].

Using this data set, we investigated how the two interaction layers are connected and the consequences for the robustness of these networks to plant loss. Robustness was assessed by sequentially removing plants in a random order and estimating secondary extinctions (Fig 1C and Methods). Although this approach lacks realism (since there are no underlying temporal dynamics), it has proven useful in understanding the threat that biodiversity loss poses to ecosystem services and functioning [3, 21, 31, 32]. Furthermore, it provides a lower bound on the damages that may be caused to an ecological community since it relies on the conservative hypothesis that secondary extinctions happen only when an animal species has lost all its links. We focused on the extinctions of plants because they are the only group of species, whose disappearance can potentially harm all other species groups, and also because plants can be managed more directly [21]. Note that while plants are not the shared set of species in all networks (see Fig 1), it is still possible to quantify robustness to plant loss in all the networks of our data set (see Methods). Extending the study of robustness to include multiple interactions simultaneously allowed us to study the interdependence of the robustness of animal species sets (Fig

[1D](), which is relevant to know how cascading extinctions will propagate through a multi-interaction network [21], and to better identify keystone plant species [13, 21], of importance when designing protection and restoration interventions. We used four null models with increasing constraints (see Methods) to study how different structural properties could determine the interdependence and robustness in the tripartite networks.

Taken together, our results suggest that considering multiple ecological interactions simultaneously does not have a dramatic impact on the robustness of tripartite networks to plant losses. However, a multi-interaction approach is crucial to better gauge the overall robustness of ecological communities, to know the interdependence of the robustness of the different animal sets, and to correctly determine the relative importance of different plants species at the whole community level, which can be key for biodiversity conservation.

## Results

### Different ways of connecting the interaction layers

We gathered a total of 44 ecological networks, each containing two types of ecological interactions, including mutualistic (pollination, seed-dispersal and ant-mutualism, corresponding to respectively 19, 3 and 1 networks) and antagonistic (herbivory and parasitism, corresponding to respectively 41 and 24 networks) interactions (See Tables A and B in S1 Text, and Figs A-D in S1 Text for more details). We divided these networks in three types according to the signs of their interactions: mutualistic-mutualistc, mutualistic-antagonistic, and antagonistic-antagonistic (see Methods).

To study how the interaction layers are connected, we focused our attention on the shared set of species between the two interaction layers. We measured three structural properties of the shared species: the proportion of the shared species that are connector nodes, i.e. that have links in both interaction layers ($C$); the proportion of shared species hubs, i.e. 20% of the shared species with the most connections, that are connectors nodes ($H_C$); and the participation coefficient of the connector nodes between the two interaction layers, i.e how well split between the two interaction layers are their links ($PC_C$) (see Methods).

This revealed fundamental differences across the three types of tripartite networks (Fig 2). In antagonistic-antagonistic networks, $\sim 35\%$ of the shared species (herbivore hosts) are involved in both parasitic and herbivory interactions (i.e. are connector nodes). Moreover, most of the shared species hubs ($\sim 96\%$) are acting as connectors between interaction layers, and they have their links equally split among the two interaction layers (average $PC_C$ of 0.89).

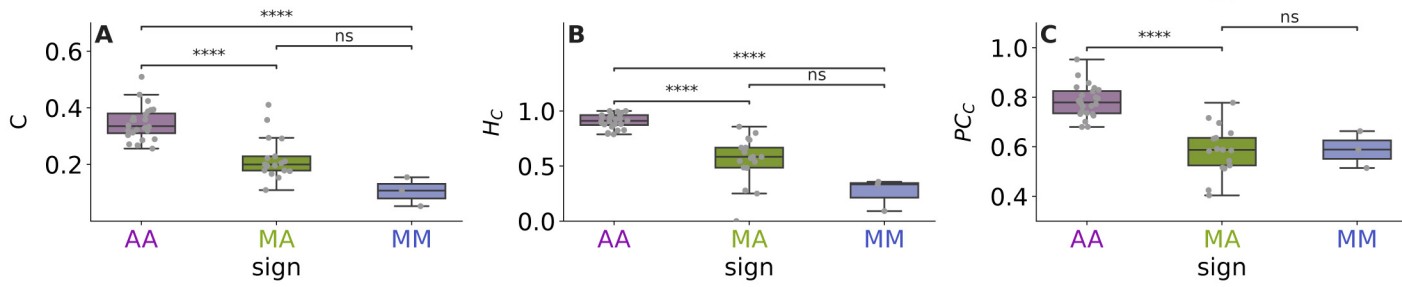

**Fig 2. How does the shared set of nodes connect the network?** A) Proportion of connector nodes in the shared set, B) Proportion of shared set hubs that are connector nodes, C) Average participation coefficient of the connector nodes. Boxplots are color-coded by network type: AA: Antagonistic-Antagonistic, MA: Mutualistic-Antagonistic, and MM: Mutualistic-Mutualistic. Differences among categories are measured by independent t-tests (**** p<1$e^{-4}$, *** p<1$e^{-3}$, *ns* not significant).

We found a very different pattern in mutualistic-mutualistic networks, for which only $\sim 10\%$ of the shared species (plants in this case) are involved simultaneously in the two types of mutualistic interactions, and only 32% of shared species hubs act as connector nodes. Also, the connector nodes have their links less equally split among the two interaction layers (average $PC_C$ of 0.59). Mutualistic-antagonistic networks are not significantly different from mutualistic-mutualistic networks and tend to have values intermediate between those of antagonistic-antagonistic and mutualistic-mutualistic networks (Fig 2A–2C). About $\sim 22\%$ of the shared species are involved simultaneously in the two types of mutualistic interactions, $\sim 56\%$ of shared species hubs act as connector nodes and the average $PC_C$ is $\sim 0.59$. An example of this contrasting structure is visible at a glance in the way the connector nodes link the interaction layers differently in the two networks in Fig 1.

## Interdependence of the robustness of animal species

We expected these differences in structure to affect the correlation of the robustness of the two animal species sets. Following recent studies [21, 28], for each network, we measured the robustness of the two animal species sets following the extinction of plants, and we investigated whether they were correlated, i.e. if they were interdependent (Fig 1D and Methods). When driving plants to extinction, a 'high' correlation between the robustness of the two animal species sets implies that the same plants that are important for one of the species set are also important in the other species set [21], (e.g. the plants whose extinctions lead to a relatively high number of secondary extinctions of pollinators also do so for herbivores).

We found that, in general, when plants are driven to extinction in a random order, interdependence ($I$) is either positive or null (Fig 3A), with, again, fundamental differences between antagonistic-antagonistic networks and the two other types of networks. The value of interdependence found in antagonistic-antagonistic networks is on average significantly higher from that found in the other two network types, which is consistent with our results on hubs and connectors, suggesting that the two layers in antagonistic-antagonistic networks tend to be strongly interconnected. Note that data collection for parasitoids relies on their sampling on herbivores found on leaves. This sampling difference (compared to the two other types of networks where species sets can be collected independently from each other) could potentially introduce a positive correlation. However, the correlation we found is not significantly higher

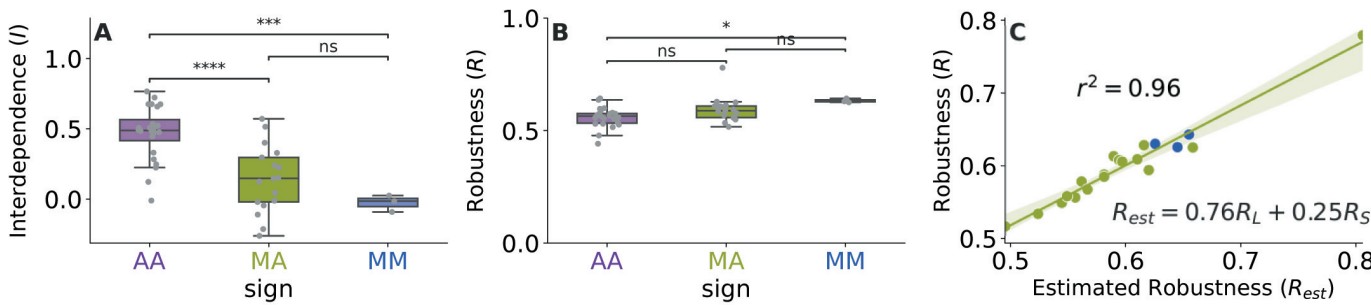

**Fig 3. Interdependence and robustness of tripartite networks.** A) Interdependence ($I$) of the tripartite networks in our data set. As $I \to 1$ the importance of plants for the maintenance of the two animal species sets becomes more similar. B) Robustness of the tripartite networks in our data set ($R$) when plants are randomly driven to extinction. As $R \to 1$, animal groups are increasingly robust to the simulated sequential loss of plant taxa. Grey points represent the values in each network. All boxplots are color-coded based on the type of tripartite network. Differences among the categories are measured by independent t-tests (**** p$<1e^{-4}$, *** p$<1e^{-3}$, * p$<5e^{-2}$, *ns* not significant). C) Robustness ($R$) vs Estimated Robustness ($R_{est}$) in the empirical MA and MM networks of our database. The text shows the best estimation of the robustness as a combination of the robustness of the larger ($R_L$) and smaller ($R_S$) bipartite networks that compose the tripartite network, and the correlation coefficient. Each point represents a network, color coded based on network type. AA: Antagonistic-Antagonistic in purple, MA: Mutualistic-Antagonistic in green, and MM: Mutualistic-Mutualistic in blue.

from what is expected in the null models. The presence of this positive correlation in the four null models considered (Fig E in S1 Text) suggests that it is due to the particular layout of these networks, more specifically, to the cascading extinction process characteristic of these tripartite antagonistic-antagonistic networks, in which plants are not the shared set of species, meaning that their extinctions sequentially spread from plants to herbivores and to parasites.

In mutualistic-mutualistic networks, the interdependence is close to null, meaning that the robustness of the two species sets seem largely decoupled from each other (but more correlated than expected by chance if we do not control for degree heterogeneity, i.e. the heterogeneity of the number of links each species has (panels B and C in Fig E in S1 Text).

Mutualistic-antagonistic networks exhibit a range of values going from moderate correlations ($I \sim 0.5$) to weak negative correlations ($I \sim -0.2$), and comparisons with null models showed a similar trend as in mutualistic-mutualistic networks, with empirical networks being more correlated than their randomized counterparts without taking degree heterogeneity into account (panels B and C in Fig E in S1 Text.).

Studying how interdependence relates to the three structural features we measured revealed differences among network types as well. More specifically, in antagonistic-antagonistic networks, interdependence is correlated (albeit weakly) with the proportion of connectors ($C$), while in the other two network types it varies with the proportion of hubs that are connectors ($H_C$) and their (un)balanced participation in the two interaction layers ($PC_C$) (Table 1, and see Fig F and Table C in S1 Text for more details).

## Tripartite networks' robustness

The robustness of antagonistic-antagonistic networks was found to be lower than that of Mutualistic-Mutualistic networks when plants were randomly driven to extinction (Fig 3B), although differences among the three types of networks are overall not significant. Surprisingly, this suggests that even if the different ways in which the tripartite networks are connected seem to have a significant effect on interdependence, this difference does not translate into significant differences in the global robustness of the tripartite networks. In other terms, a higher interdependence between the interaction layers does not cause a lower overall

**Table 1. Table of regression of interdependence ($I$) and robustness ($R$) on the structural features we studied: Degree heterogeneity ($\sigma_k / <k>$), proportion of connector nodes ($C$), proportion of shared species hubs that are connectors ($H_C$), and (un)even split of interactions among interaction layers ($PC_C$).**

|  | $I$ | | $R$ | |
| --- | --- | --- | --- | --- |
|  | **AA** | **MA & MM** | **AA** | **MA & MM** |
| $\sigma_k / <k>$ |  | 0.30 | 0.68*** | 0.38* |
| $C$ | 0.40* |  | 0.24** | -0.51** |
| $H_C$ |  | 0.70*** |  |  |
| $PC_C$ |  | 0.50** | -0.25* |  |
| Observations | 24 | 20 | 24 | 20 |
| $R^2$ | 0.16 | 0.70 | 0.76 | 0.38 |
| Adjusted $R^2$ | 0.12 | 0.64 | 0.73 | 0.31 |
| F Statistic | 4.20* | 12.41*** | 21.39*** | 5.18** |

*Note*:

*p<0.1;

**p<0.05;

***p<0.01

robustness. As expected, all tripartite ecological networks were most fragile when plants were selectively attacked targeting the most connected plants first, and the least fragile when plants were attacked selecting the specialists plants first, as previously reported in networks with only one interaction type [2, 3, 33–35] (Fig G in S1 Text).

The structural features that most determine the robustness of the networks are the degree heterogeneity and the proportion of connector nodes in mutualistic-antagonistic and in mutualistic-mutualistic networks, as well as the even split of links between the two interaction layers in antagonistic-antagonistic networks (Table 1, and see Figs H and I, and Table D in S1 Text for more details). We included the degree heterogeneity of nodes in the analysis (i.e. the variance of the interaction degree divided by the average degree) because broad degree distributions are known to make ecological networks with one interaction type more robust to random deletion of species [3, 36], a result we recover here in the case of tripartite networks. Comparison with the null models further corroborates this result, since the robustness of mutualistic-mutualistic and mutualistic-antagonistic networks was not significantly different from that of their randomized counterparts when degree heterogeneity is conserved (panels C to E in Fig J in S1 Text).

Furthermore, the robustness of the tripartite networks could be predicted by the robustness of the two bipartite networks composing it (Fig 3C). The estimated overall robustness, a combination of the robustness of the two bipartite networks (Methods), is in very good agreement ($R^2 = 0.96$) with the robustness of the tripartite networks. When the robustness of only one bipartite network was used, $R^2$ was at most 0.8 (Fig K in S1 Text). While in the main text we only consider the classical co-extinction algorithm in unweighted networks because it is the more parsimonious and offers a lower bound to the damage the community can suffer, we show that the results hold when using a stochastic version in weighted networks [37] (Figs L and M in S1 Text).

## Plant importance for robustness

The results on interdependence suggest that the important plants for one set of animal species may not always be as important for the other species set (e.g. important plants for pollinators may not be important for herbivores and vice versa). We investigated this point further and asked which plants were more important for the survival of the whole ecological community, and to what extent those plants were the same for the two animal species sets. We therefore built three rankings of plant importance—one for each animal species set and one for the whole community—in which a plant is considered to be more important if robustness is lower when that plant is attacked earlier in the extinction sequence [21] (Methods). For example, a plant can be considered important based on the pollinator and whole community rankings (e.g. plant 1, Fig 4A and 4B), but not so based on the herbivore ranking (Fig 4C). Other plants can be important based on the three rankings (e.g. plant 2, Fig 4D–4F). Comparing the three rankings in the example shows that plant importance when the two interaction layers are considered simultaneously (whole community) is not just a simple combination of the ranking of plant importance for each set of animal species (Fig 4G). While it is more difficult to differentiate between the less important plants (those with lower values of importance), the ranking is well defined, as can be seen from the correlation values between the importance and the ranking based on importance (Fig N in S1 Text). Interestingly, it becomes better defined when the two interaction layers are considered simultaneously (Fig O in S1 Text).

We studied to what extent the importance of a given plant at the whole community level was driven by its importance for the two animal species sets (Fig 4H). In the majority of networks ($\sim 63\%$), the importance of a plant for the whole community is a mixture between its

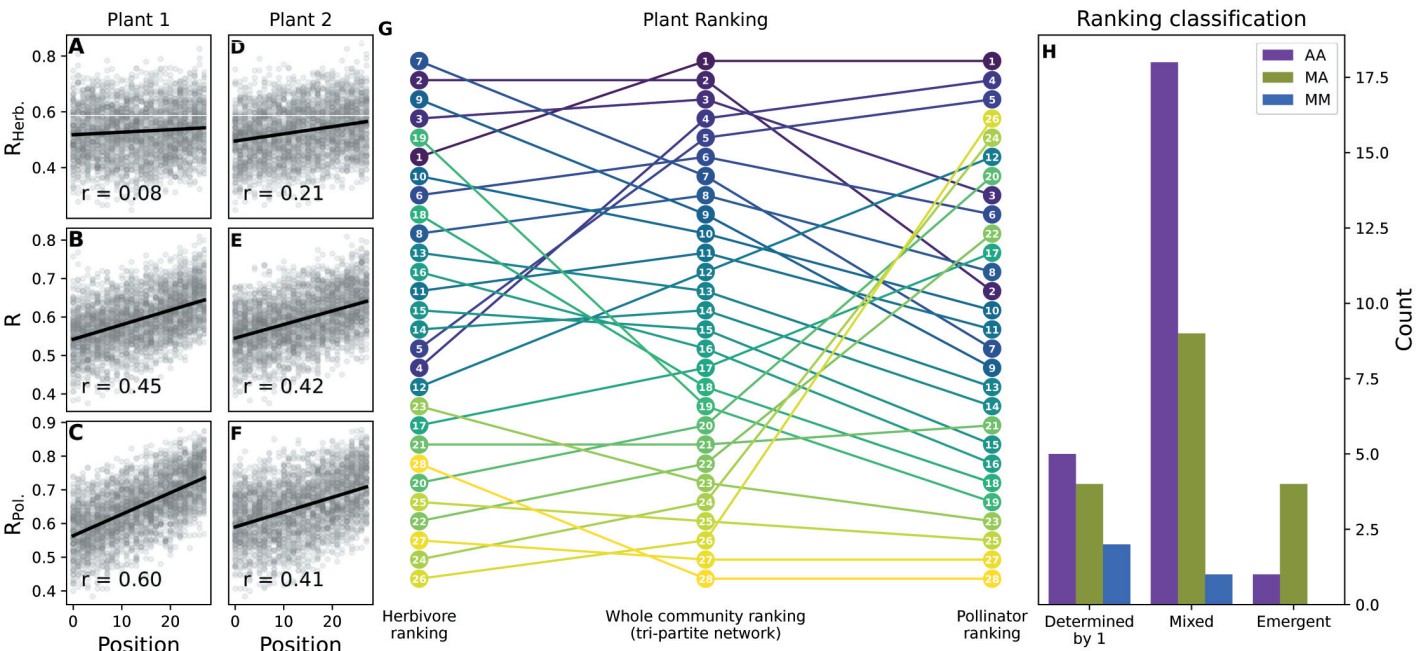

**Fig 4. Plant importance rankings.** A-F: Scatter plot of the robustness of pollinators ($R_P$), of the tripartite network ($R$), and of herbivores ($R_H$) vs the order of two plants (plant 1 and plant 2, chosen as an illustrative example) in the extinction sequence. The correlation coefficients are used to determine the ranking of importance of plant species. G: Ranking of plant importance for pollinators (right), for the whole community (center) and for herbivores (left). Each plant is represented by a disk whose number reflects its order in the ranking of importance of the whole community (in the tripartite network). The height of the disk represents its order in each of the three different rankings (i.e. the higher the position, the more important). Lines between balls are a visual help to track changes in the rankings. H: Classification of the tripartite networks in our database according to $S_{set}$ (similarity between the ranking of plant importance in the whole community and in the animal sets), illustrating if the ranking of plant importance in the whole community is mainly determined by only one animal set, is a mixture of the rankings of importance in the two animal sets (mixed), or does not resemble any of the rankings of importance in the two animal sets (emergent).

importance for the two animal sets (i.e. the similarity between the ranking in the whole community and in the animal sets ($S_{set}$) is between 0.5 and 0.9; Methods), while in $\sim 25\%$ of the networks it is mostly driven by its ranking in one of the animal species sets (i.e $S_{set}$ of one animal species set is above 0.9). This was especially relevant in mutualistic-mutualistic networks, where 2 out of the 3 networks lie in this category, probably because of the high dissimilarity between the sizes of the two animal sets (180 pollinators vs 27 seed-dispersers and 173 pollinators vs 30 ants). In a few cases ($\sim 12\%$), the ranking of plant importance for the whole community did not resemble any of the rankings for the animal sets (i.e. both $S_{set}$ were below 0.5), meaning that the importance of a plant when the two interactions are considered simultaneously changes dramatically compared to its importance when the interactions are considered separately. While in the main text we only consider the classical co-extinction algorithm in unweighted networks because it is the more parsimonious and offers a lower bound to the damage the community can suffer, we show that these results hold when using a stochastic version in weighted networks (Figs L and M in S1 Text).

## Discussion

We gathered 44 tripartite ecological networks composed by two types of ecological interactions (including herbivory, parasitism, pollination, seed dispersal, and ant-mutualism) to investigate how different interaction types were connected to each other in tripartite ecological networks and to study how considering multiple interactions simultaneously changed our knowledge of

their robustness to plant loss. While multi-interaction network data sets have been gradually appearing in the literature in the last years, only a few studies have compared several of them [13]. Such comparison allows us to reveal possible commonalities of network properties (or particularities) across the different types of tripartite networks, categorized based on the sign of the ecological interactions composing them. The rationale behind this categorization is that previous studies showed that the structure of mutualistic and antagonistic ecological networks was clearly different [30].

We found fundamental differences in the way the two interaction layers are connected in the different types of tripartite networks (Fig 2A–2C), possibly as a consequence of underlying biological constraints. In antagonistic-antagonistic networks, the shared species hubs are almost all connectors (meaning that generalist herbivores tend to have more parasitoids, maybe because they tend to be more abundant too, or maybe due to the sampling procedure in which parasitoids can only be reared out of the sampled herbivores), while in mutualistic-mutualistic networks most shared species hubs are not connectors (meaning that generalist plants tend not to be involved in two types of mutualism simultaneously, which hints at trade-offs in the type of interactions a given species can invest in, making it unlikely that a species can e.g. invest in attracting both pollinators and ant bodyguards [38]). The more varied behaviour of mutualistic-antagonistic networks may be related to highly complex trade-offs between herbivory and pollination [39].

Intuitively, we expected these differences in the connection patterns to affect the correlation between the robustness of the animal species sets in the different types of tripartite networks. These correlations (which we named 'interdependence') suggest that in antagonistic-antagonistic networks the same plant species are important for both animal sets (in terms of secondary extinctions), whereas this is not the case in mutualistic-mutualistic networks. Our results add to previous evidence showing that the benefits of an intervention are not always expected to propagate throughout the whole network [21], which has implications for biodiversity conservation. They highlight the relevance of knowing the type of ecological interactions involved in an ecological community before planning restoration efforts, since, in the analysed networks containing mutualistic interactions, positive cascading effects could only be expected if the generalist plants acted as connector nodes and were the focus of the restoration plan.

Surprisingly, we found that more interdependent communities are not necessarily less robust to plant losses. Rather, robustness of the overall tripartite network is determined by the particular organization of each network, with degree heterogeneity playing an important role, especially in antagonistic-antagonistic networks. The positive effect of degree heterogeneity on the robustness of food webs and bipartite mutualistic networks was already reported in [36] and in [3, 40] (in mutualistic networks through nestedness, but it was also shown that nestedness is a consequence of degree heterogeneity [41]). It is worth noting that the robustness of mutualistic-mutualistic and mutualistic-antagonistic tripartite networks was found to be a combination of the robustness of the two bipartite networks composing them, stressing the relevance of knowing the structure of connections in both interaction layers to better quantify the robustness of the whole tripartite network. This is good news for ecologists, because it means that when measuring overall robustness to plant loss it is still possible to use multiple bipartite networks (with only one interaction type) and assume their effects are additive, as long as we know how plants connect them. Interestingly, looking at the two interaction layers simultaneously did not result in a dramatic change in the robustness of the whole community, as already reported for one of the networks in the database [24]. Nonetheless, considering the two interactions simultaneously improved the quantification of the overall robustness and is crucial to identify the most important plants in a given community.

The approach we used to study robustness also allowed us to identify keystone species in the whole community. In most tripartite networks, the ranking of plant importance in the whole community is determined by the importance of plants for both animal sets (with the exception of mutualistic-mutualistic networks, that are mostly driven by one interaction layer, probably because of their disproportionate size and low connection among interaction layers). In a few cases, considering the whole community could even alter the picture considerably, since the ranking of plant importance in the whole community is emergent, i.e. it is not similar to the ranking of importance for neither of the animal sets. This evidences that considering multiple interactions simultaneously can be crucial for correctly identifying keystone species in a community.

The results we present here advance our knowledge of how different interactions connect ecological communities, and how that affects the robustness of tripartite networks to plant losses. Taken together, our results suggest that considering multiple ecological interactions simultaneously does not have a dramatic impact on the overall robustness of multi-interaction networks to plant losses. However, a multi-interaction approach is crucial to know the interdependence of the robustness of the different animal sets, to better gauge the overall robustness, and to correctly determine the importance of the plants at the whole community level.

## Methods

### Data set

We gathered from the literature ecological networks which included different types of interactions. Because most studies only provided two interactions simultaneously, we decided to study networks with two interaction layers. Also, we only considered unweighted networks because not all studies provided interaction strengths. From all the networks we found, we only kept those which had at least 5 connector nodes. In the end, our data set contains 44 unweighted networks from 6 studies (see Table 2). Each network is composed of two ecological bipartite layers including mutualistic (pollination, seed-dispersal and ant-mutualism) and antagonistic interactions (herbivory and parasitism). In the cases where multiple types of herbivory were present, all interactions were combined in a single herbivory layer. See Tables A and B in S1 Text, and Fig A in S1 Text for more details.

### Structural metrics of the connector nodes

We were interested in studying how the two different interactions of the tripartite networks were interconnected through the connector nodes. We used three metrics to quantify this:

- The proportion of connectors nodes in the shared set of species ($C$), i.e. the proportion of shared species that have links simultaneously in the two interaction layers [25].

**Table 2. Tripartite networks included in our analyses, indicating the sign of the interactions (i.e. if the tripartite network has both mutualistic and antagonistic interactions (MA), only antagonistic interactions (AA), or only mutualistic interactions (MM)), the two ecological interactions composing the tripartite network, the number of network of each type, and the reference.**

| Sign | Interactions (Acronym) | Number of networks | references |
|------|------------------------|--------------------|------------|
| MA | herbivory-pollination (H-P) | 16 | [6] [42] [21] [43] |
|    | herbivory-seed dispersal (H-SD) | 1 | [6] |
| AA | herbivory-parasitism (H-Pa) | 24 | [44] [43] |
| MM | pollination-seed dispersal (P-SD) | 2 | [24] [6] |
|    | pollination-ant mutualism (P-A) | 1 | [24] |

- The proportion of shared species hubs that are connectors ($H_C$), i.e. the 20% of the species in the shared set of species with the highest degree that are connector nodes. Note that the degree of a node is the number of links it has with other species. We used a threshold of 20% to ensure that all networks had at least 1 "most connected" node, but the results are robust to that choice (Fig P in S1 Text).

- The participation coefficient. This species-level metric quantifies whether the links of node $i$ are primarily concentrated in one interaction layer or if they are well distributed among the two interaction layers [45, 46]. We quantified it as two times the ratio between the lowest degree in both interaction layers divided by the total degree of the node ($2\frac{k_{min}}{k_{tot}}$). Hence $PC = 1$ if the links are perfectly split among the two interaction layers, and it approaches 0 as the split grows more uneven. We obtained the participation coefficient of the connector nodes ($PC_C$) by computing the average value over the connector nodes.

## Quantifying robustness

We simulated plant loss following an established method [2, 3] and assuming bottom-up control of the animals, as justified by [21, 47]. To quantify robustness to plant loss we sequentially removed plants in a given order (the 'extinction sequence') keeping track of the number of secondary extinctions of animal species at each step. We considered that an animal species undergoes extinction when it has lost all its links. Note that secondary extinctions work differently in mutualistic-antagonistic and mutualistic-mutualistic networks compared to antagonistic-antagonistic networks. In the former, after removing a plant, all herbivores that no longer have resources go extinct and so do all pollinators without any resources, which means that erasing a plant may generate *simultaneous* secondary extinctions in the two animal species set (Fig 1A). In antagonistic-antagonistic networks herbivores are the shared set of species, so when a plant disappears all herbivores without resources go extinct, which may subsequently trigger extinctions of parasitoids. In this case, removal of a plant will generate *cascading* extinctions (Fig 1B). By plotting the proportion of remaining animal species as a function of the proportion of deleted plant species and measuring the area under the curve, we obtained the 'robustness' ($R$) (Fig 1C). This is a standard way of measuring the efficiency of a given extinction sequence to tear down an ecological community [48, 49]: as $R \rightarrow 0$ the most impact a given extinction sequence has on the community, indicating that it targets the species following the 'correct' order of importance for the maintenance of the community.

Working with multipartite networks such as those in [21, 24], several robustness metrics can be measured depending on the species set on which secondary extinctions are considered. Here, we measured:

- the robustness of the tripartite network ($R$): we kept track of the proportion of remaining animal species as a function of the proportion of deleted plant species, where the proportions are measured with respect to the total number of animals (irrespective of their species set) and plants.

- the robustness of the two animal species sets ($R_P$, $R_H$): we measured the proportion of remaining animal species with respect to the total number of animals in each species set (e.g. how many pollinators remain from the original number of pollinators), and the proportion of deleted plants is measured with respect to the total number of plants in the tripartite network.

- the robustness of the two bipartite networks: In this case, the tripartite network is split in two bipartite networks, on which the same protocol as above is performed. These two

networks are not identical to the two interaction layers because the shared set of species that are not connected in a given layer are not considered in the bipartite network, which affects the calculation of the robustness. We thereby obtain two robustness ($R_L$ and $R_S$, respectively for the smaller and larger networks, in terms of species number). Note that in antagonistic-antagonistic networks, the protocol can be performed only on the herbivory network since there is no direct link between plants and parasitoids.

We applied 3000 random extinction sequences of plants to each of the tripartite networks in the data set, and for each extinction sequence we measured the different robustness measures above. Here, results are presented for random extinction sequences but results for other extinction scenarios (increasing or decreasing degree of plants) are presented in Figs G, H and I in S1 Text.

We also measured the robustness of the mutualistic-antagonistic network using an stochastic version of the co-extinction algorithm [37] and weighted networks (when available) to compare with the results of the classic co-extinction algorithm (see Figs L and M in S1 Text). In this stochastic version, a species $i$ will undergo a secondary extinction following the extinction of plant $j$ with a probability $P_{ij} = R_i.d_{ij}$, where $d_{ij}$ is the dependency of species $i$ on $j$ (interaction weight), and $R_i$ represents the intrinsic demographic dependence of species $i$ on mutualism (we considered $R_i = 1$ for animals and $R_i = 0$ for plants, to keep the bottom-up control of animals).

## Interdependence

We measured the correlation between the robustness of the two different species sets (other than plants) in the tripartite networks, hereafter called 'interdepence' (*I*) (Fig 1D). When driving plants to extinction, a 'high' correlation between the robustness of pollinators and herbivores implies that the same plants that are relevant for one of the species set will also be relevant in the other species set [21], (i.e. sequences of plant loss that were relatively benign for pollinators were also benign for herbivores). If, on the other hand, the relevant plants are not the same in the two species sets, we expect a low correlation in robustness.

## Plant importance rankings

The importance of each plant species for the different animal sets and for the whole community (i.e. for the tripartite network) was quantified based on the correlation coefficient between robustness and the position of the plant in the extinction sequence [21]. The rationale is that the 'importance' of a plant cannot be directly assessed from the number of secondary extinctions caused by its loss because if lost at the start (rather than at the end) of the extinction sequence, fewer secondary extinctions are expected; however, if a plant is 'important', then robustness is expected to be lower when it is lost earlier in the sequence than when it is lost later. Hence, the lower the robustness to an extinction sequence, the better that extinction sequence actually resembles the importance of plants for the survival of the community. To obtain the plant importance rankings (three in total: one for each of the two interaction layers and one for the whole community), we ranked each plant species by increasing correlation between its order of appearance in extinction sequences and the corresponding robustness (i.e. plants that have a larger negative correlation are considered more important; Fig 4A–4G).

To asses to what extent one of the two interaction layers was driving the robustness of the whole community we measured the similarity between the importance of a plant for one animal set (for example for pollinators or herbivores) and the importance of a plant in the whole

community, namely $S_P$ or $S_H$. We quantified $S_{set}$ as the square of the correlation coefficient between the ranking of the plant in each species set and the ranking in the whole community.

We then classified the networks in three categories: those where one interaction layer was driving the process (one $S_{set}$ was above 0.9, meaning that 90% of the variance in the importance ranking in the whole community can be traced to one of the two animal rankings), those where the ranking in the whole community was a mixture of the rankings in both animal sets (both $S_{set}$ were between 0.5 and 0.9) and finally those where the importance ranking was emergent (both $S_{set}$ were below 0.5, meaning that no animal set ranking was able to explain at least 50% of the ranking of importance in the whole community).

### Estimating tripartite robustness from networks with one interaction

We also tested whether one can express the robustness of the whole community ($R$) as a combination of the robustness of the two independent bipartite network composing the tripartite network. To do that we performed the following linear regression:

$$R^{(est)} = a.R_L + b.R_S$$

where $R_L$ and $R_S$ are the robustness of the two bipartite networks composing the tripartite network under study, respectively the larger (i.e. with more species) and the smaller one.

### Multiple regressions

We performed a multiple regression of interdependence and robustness based on the structural features we measured in the tripartite networks using the package *statsmodel* in Python. We selected the structural features that were more relevant for interdependence or robustness by choosing the model with a lowest AIC.

### Null models

To assess the importance of network structure in determining a certain network feature, we compared measurements of that feature performed on empirical networks with measurements performed on randomized versions of those networks keeping some properties fixed. We used four different null-models, represented in Fig 5, which—going from the least to the most constraining—are as follows: "1" keeps the number of species constant in each species set and the

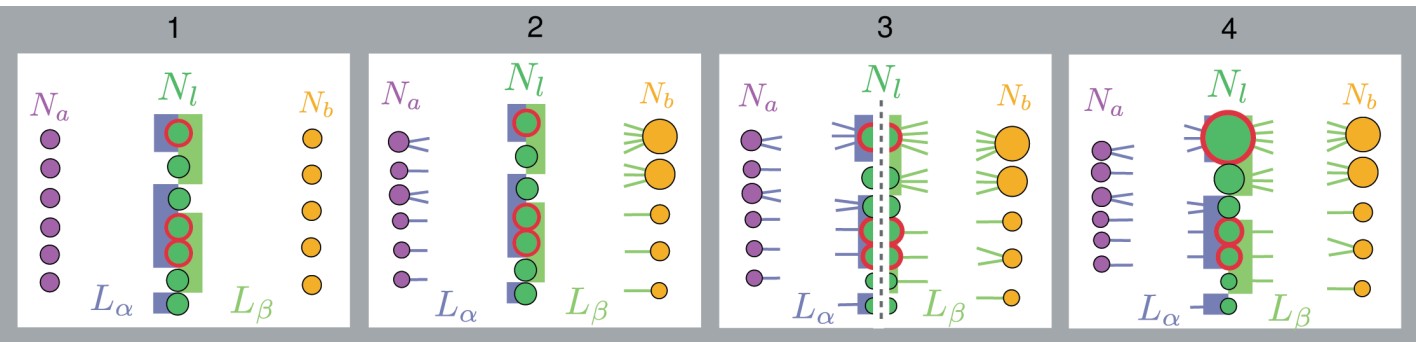

**Fig 5. The 4 different null models used in this study.** Each figure represents what is kept fixed in each null model, going from the less restrictive on the left, to the more restrictive on the right. $N_x$ is the number of nodes in the species set, $L_x$ the number of links in the interaction layer, the color of the nodes represent the different species set, the colour of the link the two different ecological interactions, the size of the node is proportional to its degree (when kept), and connector nodes are highlighted in red.

number of links constant in each interaction layer, "2" adds the constraint of keeping the degree distribution of the animal nodes constant, "3" keeps the degree distribution of animals and plants but not the total degree of the shared set species (i.e. it breaks the correlation between the degree of the shared set of species in the two interaction layers), and "4" keeps the degree of each node constant while links are reshuffled within a layer (see S1 Text for more details).

## Supporting information

**S1 Text. Supplementary information file, including supplementary figures A-P and supplementary tables A-D.**
(PDF)

## Author Contributions

**Conceptualization:** Virginia Domínguez-García, Sonia Kéfi.

**Data curation:** Virginia Domínguez-García.

**Formal analysis:** Virginia Domínguez-García.

**Funding acquisition:** Sonia Kéfi.

**Investigation:** Virginia Domínguez-García, Sonia Kéfi.

**Software:** Virginia Domínguez-García.

**Supervision:** Sonia Kéfi.

**Visualization:** Virginia Domínguez-García.

**Writing – original draft:** Virginia Domínguez-García, Sonia Kéfi.

**Writing – review & editing:** Virginia Domínguez-García, Sonia Kéfi.

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
