## [Decision Letter · Decision Letter 0]

24 Mar 2023

Dear Dr Domínguez-García,

Thank you very much for submitting your manuscript "The structure and robustness of ecological networks with two interaction types." for consideration at PLOS Computational Biology.

As with all papers reviewed by the journal, your manuscript was reviewed by members of the editorial board and by independent reviewers. The three reviewers agree on the relevance and quality of the work, and have provided a series of constructive comments with questions on terminology and approach, in particular on the null models and on specifics of the robustness analysis. In light of the reviews (below this email), we would like to invite resubmission of a revised version that takes into account the reviewers' comments. 

We cannot make any decision about publication until we have seen the revised manuscript and your response to the reviewers' comments. Your revised manuscript is likely to be sent to reviewers for further evaluation.

Sincerely,

Mercedes Pascual

Academic Editor

PLOS Computational Biology

James O'Dwyer

Section Editor

PLOS Computational Biology

Reviewer's Responses to Questions

**Comments to the Authors:**

Reviewer #1: In this paper the authors investigate the robustness of tripartite networks with two interaction types. They take the most naive approach of plant removal and consequent cascading or simultaneous co-extinction of animals in the two layers. This approach has many assumptions and limitations (e.g., no rewiring, no weights). However, as the authors claim, it serves as a starting point for basic insights into community robustness. I agree with that notion. The robustness of mulitpartite networks has already been investigated. However, this is the first work to do so in a systematic manner, and comparing three types of networks. In that sense this work is novel and is a necessary contribution to the knowledge on community robustness.

There are several conceptual insights arising from this work. First, that the robustness of the tripartite networks can be predicted from that of its two constituting layers. Second, the proportion and role of connector species in the different kinds of networks. Third, that interdependent communities were not less robust to plant losses. The analysis is strong and covers multiple aspects of the structure-stability (or structure-robustness) link. The comparison to null models makes the analysis more robust and insightful, although this part is not as emphasized in the main text (see my comments). Overall, I find this paper merits publication in PLOS Comp Biol.

I have just a few comments and suggestions.

Thanks for the efforts put into that work.

The results presented in the main text are for random removal order. This is very null-model-like. It would be good to include some of the DD and ID results in the main text. Fig S9 states quite the obvious, so maybe not this one. But are there any meaningful insights regarding the effect of structural properties on robustness for nonrandom removal?

The null models provide important insights but are hidden in the SI. I really liked the approach of increasing constraints and the fiure that describes the null models. Consider moving them to the main text. I especially liked the results of interdependence that are in the SI (paragraph starting with: “Comparison with the four null models indicates that, for AA networks…”). Also Fig. S7, and S11.

Plant importance ranking: What are the actual differences between the rankings? A plant can rank higher than another but the actual difference in importance value can be very small. What is the distribution of importance values?

The inherent difference between AA and MA/MM in the role of plants will cause a "slower" (i.e., more removal events needed) extinction in AA networks. Generally, if you take a network and remove nodes from the linking set it will collapse faster than if you remove from any of the other two sets. Hence, given an AA and an MM/MA network with a similar number of nodes in corresponsing sets and similar degree distributions, AA network will be more robust than MM and MA networks. Is this the underlying cause for the differences between AA/MM-MA networks? In Fig 3B you find a pattern opposite to what I expected, whereby AA networks are less robust than MM. Why? I somehow feel that you might have explained all that in the text and I missed it…

Related — L 134: Isn’t the robustness of AA is lower than MM? At least as far as I can tell from Fig 3B.

It was difficult to understand the results without referring to the methods to see the definitions of the structural properties. Maybe move some text from the methods to the results.

How sensitive are the results for the thresholds? E.g., the 20% of the .

L108: change ‘behaviour’ to ‘structure’

Fig. 3: What are the gray data points in panels A-B? What are the green and blue points in C?

Reviewer #2: General assessment:

This is a study on the robustness of multilayer networks, using as a model different kinds of tripartite networks that combine mutualisms and antagonisms. Overall, its rationale is very interesting and the topic, timely. As claimed by the authors, multilayer networks are a new frontier in Ecology, as this approach finally provided us with a tool set to address two or more interaction types together in the same system. This is a great leap forward, as interaction types do not occur isolated in nature, but rather interact with one another, generating conditional outcomes. Tripartite networks are indeed a good starting point to advance this approach also to the assess the topic of stability. The study is well designed and well presented. Its language is strongly focused on the analyses and math, which would be problematic for a strictly biological journal, but is not the case in a more computational venue. After a few adjustments this study can become a highly relevant contribution to the fields of network ecology and species interactions.

Specific comments:

1. Be careful with typos and grammar errors, especially errors of number agreement. A review by a professional proofreader (native speaker or not) is always a good idea.

2. Avoid using so many acronyms and abbreviations in the text, especially custom-made ones that are not regularly used in the field. They may shorten the text, but severely hinder its readability.

3. A central point when studying multilayer networks is the definition of the interlayer edges. Paying attention to them is the main difference between using a monolayer or multilayer approach. In the case of tripartite networks, which fit the category of "diagonally coupled", interlayer edges occur between shared species, which are present in both layers (plants, for that matter). It is important to explain those definitions more clearly and how they affect the interdependence between layers. See:

Mucha, P. J., Richardson, T., Macon, K., Porter, M. A., & Onnela, J.-P. (2010). Community Structure in Time-Dependent, Multiscale, and Multiplex Networks. Science, 328(5980), 876 LP – 878. https://doi.org/10.1126/science.1184819

Pilosof, S., Porter, M. A., Pascual, M., & Kéfi, S. (2017). The multilayer nature of ecological networks. Nature Ecology & Evolution, 1(4), 0101. https://doi.org/10.1038/s41559-017-0101

4. Please explain more clearly how you defined the "connector nodes" and "connector hubs". There is much confusion in terminology between graph theory and network theory, also when it comes to centrality. When network ecology, this confusion becomes a real mess. For instance, in mathematical studies about multilayer networks, a node that connects two or more layers is usually named a "state node". When two nodes are connected to one another in two or more layers, their connection is named a "multilink". In the field of ecological networks, those kinds of nodes and links receive other nicknames. Furthermore, connectors and hubs are usually defined in ecological studies according to two centrality metrics: participation coefficient and within-module degree. See:

Guimerà, R., & Amaral, L. A. N. (2005). Cartography of complex networks: modules and universal roles. Journal of Statistical Mechanics: Theory and Experiment, P02001. https://doi.org/doi:10.1088/1742-5468/2005/02/P02001

Olesen, J. M., Bascompte, J., Dupont, Y. L., & Jordano, P. (2007). The modularity of pollination networks. Proceedings of the National Academy of Sciences, 104(50), 19891–19896. https://doi.org/10.1073/pnas.0706375104

Boccaletti, S., Bianconi, G., Criado, R., del Genio, C. I., Gómez-Gardeñes, J., Romance, M., Sendiña-Nadal, I., Wang, Z., & Zanin, M. (2014). The structure and dynamics of multilayer networks. Physics Reports, 544(1), 1–122. https://doi.org/10.1016/j.physrep.2014.07.001

Kivela, M., Arenas, A., Barthelemy, M., Gleeson, J. P., Moreno, Y., & Porter, M. A. (2014). Multilayer networks. Journal of Complex Networks, 2(3), 203–271. https://doi.org/10.1093/comnet/cnu016

5. In this study, the classical “Memmott approach” (or Barabasi-Albert) was used to assess secondary extinctions. As commented by the authors themselves, this approach is very simplistic and unrealistic, although it was a good first step in the early 2000s, when studies on the robustness of ecological networks began to appear. Nevertheless, much more realistic approaches have been proposed in recent years, which should be also considered. See:

Vieira, M. C., & Almeida-Neto, M. (2015). A simple stochastic model for complex coextinctions in mutualistic networks: robustness decreases with connectance. Ecology Letters, 18(2), 144–152. https://doi.org/10.1111/ele.12394

6. The authors proposed some new null models to investigate the topology of tripartite networks. This is a very nice step forward in the field of multilayer networks, especially in Ecology, where null model analysis is virtually considered mandatory. However, caution is advised. Recent studies have pointed out that some null models have been used wrongly in ecological studies. In spite of removing some undesired processes from network assembly, they might be actually simulating those same processes. This is the case of some of the null models used in this study, which were derived from classical null models proposed by Bascompte and Vázquez, among others. See:

Dormann, C. F., Frund, J., Bluthgen, N., & Gruber, B. (2009). Indices, Graphs and Null Models: Analyzing Bipartite Ecological Networks. The Open Ecology Journal, 2(1), 7–24. https://doi.org/10.2174/1874213000902010007

Farine, D. R. (2017). A guide to null models for animal social network analysis. Methods in Ecology and Evolution, 8(10), 1309–1320. https://doi.org/10.1111/2041-210X.12772

Pinheiro, R. B. P., Dormann, C. F., Felix, G. M., & Mello, M. A. R. (2021). A novel perspective on the meaning of nestedness with conceptual and methodological solutions. BioRxiv, 2021.04.05.438470. https://doi.org/10.1101/2021.04.05.438470

7. Isn’t there a better way to visualize tripartite networks? Yes, on the one hand, they are traditionally plotted as two bipartite graphs connected to one another. On the other hand, those “bipartite drawings” are quite poor in depicting topology and centrality. They serve only the purpose of visualizing nestedness. Consider more efficient, imaginative designs. See:

Pocock, M. J. O., Evans, D. M., Fontaine, C., Harvey, M., Julliard, R., McLaughlin, Ó., Silvertown, J., Tamaddoni-Nezhad, A., White, P. C. L., & Bohan, D. A. (2016). The Visualisation of Ecological Networks, and Their Use as a Tool for Engagement, Advocacy and Management. In G. Woodward & D. A. Bohan (Eds.), Advances in Ecological Research (1st ed., pp. 41–85). Academic Press. https://doi.org/10.1016/bs.aecr.2015.10.006

Marai, G. E., Pinaud, B., Bühler, K., Lex, A., & Morris, J. H. (2019). Ten simple rules to create biological network figures for communication. PLOS Computational Biology, 15(9), e1007244. https://doi.org/10.1371/journal.pcbi.1007244

Reviewer #3: Overview: I reviewed this manuscript with great interest. As the authors point out, it is only recently that datasets of networks comprised of multiple interaction types have been published in high enough numbers to start drawing conclusions across them. Here, the authors examine networks consisting of multiple antagonistic interactions, multiple mutualistic interactions and a combination of mutualistic and antagonistic interactions. They investigate structural differences and similarities as well as how robustness changes among these types of networks, and the possibility of emergent properties in the whole network not captured in the component webs. The analyses are for the most part sound and conclusions are broadly well-supported. However, I do have some questions about the effects of some analytical decisions (detailed below). For instance, it would be helpful for the authors to comment on how collection/rearing methods might be contributing to the conclusions. Also, how the decision to use unweighted networks might affect the outcome; a lot of information is lost by removing interaction strength so perhaps the authors can comment on how they might expect their conclusions to change with weighted networks? Some more technical aspects of the network metrics and descriptions could be explained earlier and in more detail (e.g. degree, connectors and hubs) to help non-specialist understanding. It is generally well written (although I have included some typos and suggestions for rewording a few awkward phrases at the end). I detail some further questions and suggestions for improvement below, but in general, I found it to be an interesting read that presents novel cross-study conclusions.

General comments:

I tend to think of tripartite networks to mean three sequentially linked trophic levels (e.g. plant-herbivore-parasitoid, or plant-herbivore-consumer), rather than combining networks of 2 interaction types that share a common resource (e.g. plant-pollinator + plant-herbivore). There’s nothing inherently incorrect with your definition, but to me it confuses the terminology a bit in a field that already has multiple terms for the same thing. Please consider using another term to describe combining networks of different interaction types. Annoyingly, the field is far from unanimous in what to call this type of network, but perhaps consider something like: network of networks, network of multiple interaction types, multilayer network, combined network, hybrid network...

A paper that might have been missed that also examined robustness across networks of different interaction types is Morrison et al. (2020). I am not suggesting this be added to the analyses per say, but it might be of interest to the authors.

Specific Comments

Line 76: In what order were species removed? Random? By degree? Can you clarify here?

Line 85: Your methods would only highlight keystone plant species, correct? Not a particularly well connected/central insect?

Lines 87-91: It reads a bit strangely to have the conclusion of the results in the introduction before having presented the results. Consider moving this to the discussion, and rather in this paragraph, outlining specifically which hypotheses you were testing/investigating and predictions.

Line 92: It would really help with readability to first have a paragraph summarising the datasets used and key properties of them (interaction types, size etc.).

Line 95: The Connector is a node that exists in both networks, correct? Not necessarily one with a particular property (e.g. high betweeness)? This could be made clearer at this point to ease understanding

Line 100: How many linking set hubs were there?

Line 119-120: How much of the interconnection in AA networks is due to it being parasitoids reared out of herbivores? Leaves with herbivores are collected and parasitoids are coming from them so they will, by necessity, be strongly interconnected. It is not necessarily the antagonism but the specific mechanism or sampling design.

Line 165: How much does this trend for dissimilarity between the sizes of networks affect your other findings? I suspect some of the MA networks were also pollinator dominated, no? Could this have skewed more of your findings?

Line 183: Can you test for the effect of abundance? Or control for it in your analyses?

Line 206-7: Your analyses don't account for any species that might exist in multiple networks (e.g. butterflies). Can you comment on how this might affect your conclusions? Could the impact of looking at robustness in combined networks be stronger if this were included?

Line 224: Using unweighted networks means potentially important information is lost when removing interaction strength. Can you comment on how this might have affected your conclusions? Was there any attempt to normalise across networks of different sizes?

Line 235: Non-specialists might not know what degree is. Probably worth defining on first use.

Line 243: How was the order of removal determined? From least connected/lowest degree to most? randomly? another way?

Line 256: Multipartite networks is inconsistent terminology. I say above about my preference to not use tripartite here, but whatever term is used, it should be consistent

Line 271: So the extinction sequence is based on random removal? Can you clarify this earlier

Line 281-2: This is a neat approach. Does it correlate at all with degree? Did you get the same/similar importance when species were removed in order of degree?

SI table 1: How did you handle multiple types of herbivory (e.g. leaf miners, caterpillars)? Were they combined into 1 network? Or did you not use networks with combined herbivory (although I think some of the studies did have multiple herbivores, no?)

Figures:

Fig. 2 caption: Please define PR in the caption for clarity.

The asterisks seems 1 too many. I tend to think of * for < 0.05, ** for < 0.01, *** for < 0.001

Minor issues (e.g. typos, syntax etc.):

Line 27: “trough” should be “through”

Line 100: “the ones doing the connection” is awkward wording. Consider something along the lines of “…are acting as connectors” or “are forming connections”

SI 2: 1st paragraph “less” and “more” should be “least” and “most”, respectively

In NL2 paragraph (and other places): “allows to study” is not correct wording. Replace with “allows the study of” or “allows us to study” or similar.

References:

Morrison, B. M. L., Brosi, B. J. & Dirzo, R. Agricultural intensification drives changes in hybrid network robustness by modifying network structure. Ecology Letters 23, 359–369 (2020).

**Have the authors made all data and (if applicable) computational code underlying the findings in their manuscript fully available?**

Reviewer #1: Yes

Reviewer #2: Yes

Reviewer #3: None

PLOS authors have the option to publish the peer review history of their article (what does this mean?). If published, this will include your full peer review and any attached files.

Reviewer #1: No

Reviewer #2: No

Reviewer #3: No
---

## [Decision Letter · Decision Letter 1]

26 Oct 2023

Dear Dr Domínguez-García,

Thank you very much for submitting your manuscript "The structure and robustness of ecological networks with two interaction types." for consideration at PLOS Computational Biology. As with all papers reviewed by the journal, your manuscript was reviewed by members of the editorial board and by several independent reviewers. 

All reviewers found the revised manuscript clearer and more compelling. Based on the reviews, we are very likely to accept this manuscript for publication, providing that you modify the manuscript to incorporate the discussion point suggested by reviewer 3 and address the point on the data made by reviewer 1.

Sincerely,

Mercedes Pascual

Academic Editor

PLOS Computational Biology

James O'Dwyer

Section Editor

PLOS Computational Biology

Reviewer's Responses to Questions

**Comments to the Authors:**

Reviewer #1: The authors have addressed my comments. My only comment is that they should remove line 69-73 (starting with: "In the network science literature..."). This statement is wrong. Diagonally-coupled networks have interlayer links. The tripartite networks are not multilayer and do not have interlayer links. They are multipartite (tripartite in this case).

Reviewer #2: The authors did a terrific job revising their manuscript. This new version is much better than the previous. I have no further suggestions to make.

Reviewer #3: The authors have done a great job of addressing the comments and this revised version is much improved. It is clearer to read and the questions about the methodology have been fully addressed and the terminology and approach is much clearer throughout.

The main point that remains, to my mind, is with regard to the relevance of the comparisons between different types of tipartite networks. Because the original data sets were not necessarily collected to compare between the different bipartite networks that comprise them, some of the detected differences might be due to sampling differences rather than structural ones. In particular, the antagonistic-antagonistic networks are all plant-herbivore-parasitoid networks which is a specific kind of A-A network (as supposed to say, herbivory and insectivory combinations that could be collected independently). These are typically collected by sampling a herbivore on a plant and then rearing a parasitoid out of it. Whereas a plant-pollinator/plant-herbivore or plant-pollinator/plant-seed disperser network would have 2 independent measures for each interaction. This means that the herbivore linking to a parasitoid is the only way a parasitoid could be in the network. The authors now note that this is relevant for the robustness analyses as any plant removal simulation would necessarily propagate up the A-A network due to the sampling design (but also biologically) rather than any network property (a parasitoid has no ability to interact with a plant outside of the sampled herbivores). But I think it is important for the “fundamental differences” reported as well.

I’m not sure there is a way to disentangle this confounding effect, and the conclusions are still relevant and interesting, but it is a consideration that needs to be acknowledged, particularly in the discussion.

Specifically, in lines 210-212: the conclusion that the hubs being connectors is because generalist herbivores might be more abundant, could also be because the parasitoid can only be reared out of one of the sampled herbivores. I would be very cautious about the interpretation here being about an ecological or structural difference when it could be an unavoidable sampling effect.

The same is true in the respective results (lines 113-114) regarding the equal splitting of links. Sampling methods mean the herbivore hub would necessarily be equally split between the 2 layers. If the links were collected independently (as in plant-pollinator/plant-herbivore networks), then you might expect fewer shared links. It is important to distinguish which of the results are due to collection method differences, especially since the data were typically not collected to test for these sorts of differences so the variable collection methods are likely to be an important confounding effects.

Otherwise, the conclusions are well supported by the results and the approach including multiple null models and now including weighted networks, where possible, is sound and much improved!

Very minor typo/syntax suggestions:

Line 74: “consists in 44…” should be “consists of 44…”

**Have the authors made all data and (if applicable) computational code underlying the findings in their manuscript fully available?**

Reviewer #1: **No: **I did not see a link to the data the authors used. I might have missed it, but it must appear. Referring to the original papers is not good enough because sometimes there are changes when using other peoples data. The authors should store their data online in a permanent data repository (Figshare, dryad, etc).

Reviewer #2: Yes

Reviewer #3: None

PLOS authors have the option to publish the peer review history of their article (what does this mean?). If published, this will include your full peer review and any attached files.

Reviewer #1: No

Reviewer #2: No

Reviewer #3: No

Figure Files:

Data Requirements:

Reproducibility:

References:

---

## [Editor Report · Decision Letter 2]

18 Dec 2023

Dear Dr Domínguez-García,

We are pleased to inform you that your manuscript 'The structure and robustness of ecological networks with two interaction types' has been provisionally accepted for publication in PLOS Computational Biology.

Best regards,

Mercedes Pascual

Academic Editor

PLOS Computational Biology

James O'Dwyer

Section Editor

PLOS Computational Biology

---

## [Editor Report · Acceptance letter]

15 Jan 2024

PCOMPBIOL-D-23-00025R2 

The structure and robustness of ecological networks with two interaction types

Dear Dr Domínguez-García,

I am pleased to inform you that your manuscript has been formally accepted for publication in PLOS Computational Biology. Your manuscript is now with our production department and you will be notified of the publication date in due course.

With kind regards,

Zsofi Zombor
